# Impact of monitor unit optimization in volumetric modulated arc therapy planning for nasopharyngeal carcinoma

**Huaqu Zeng[1]\*, Zhen Li[2], Zongyou Chen[1], Shukui Tang[1], Qifu Lin[1], Minzhi Zhong[3], Zunbei Wen[1]\***

**1** Radiotherapy Center, Gaozhou People's Hospital, Gaozhou, Guangdong, China, **2** Department of Oncology Radiotherapy, Zhongshan City People's Hospital, Zhongshan, Guangdong, China, **3** Department of Radiology, Guangzhou Red Cross Hospital, Guangzhou, Guangdong, China

\* 756660752@qq.com (HZ); zbwen@139.com (ZW)

## Abstract

### Purpose

To evaluate the impact of monitor units (MUs) optimization on volumetric modulated arc therapy (VMAT) plan for nasopharyngeal carcinoma (NPC).

### Methods

Twenty-one NPC patients were retrospectively analyzed. Dual-arc VMAT plan were designed using photon optimization algorithms without the monitor unit objective (MUO) tool, denoted as the base plan. Each base plan was re-optimized with the MUO tool with the Maximum MU parameter set to 30% of the base plans' total MUs and Strength parameters set to 50, 80, and 100, generating plans $S_{50}$, $S_{80}$, and $S_{100}$. Target and organ-at-risk (OAR) dose distributions, MUs, beam delivery time, and gamma passing rates were compared between re-optimized and base plans. Statistical analysis was performed using SPSS 17.0 (paired t-tests; significance: $P < 0.05$).

### Results

Plan $S_{100}$ reduced target PCTV2 $D_{98\%}$ by >4% (relative to the base plan) in four patients. Plan $S_{80}$ reduced target PGTV and PGTVnd $D_{max}$ and target PCTV2 $D_{98\%}$ for >3% but <4% in two patients, while other target dose parameters changed by <2%. Compared to the base plan, all re-optimized plans increased the brainstem $D_{max}$ ($P < 0.05$), though the maximum increase was <1.5%. Plan $S_{50}$ reduced both parotid glands $D_{50\%}$ and $D_{mean}$ ($P < 0.001$), while plan $S_{80}$ reduced both parotids $D_{mean}$ and the left parotid $D_{50\%}$ ($P < 0.001$). Conversely, $S_{100}$ increased both parotids $D_{50\%}$ and $D_{mean}$ and the spinal cord $D_{max}$ ($P < 0.05$). Plan $S_{80}$ and $S_{100}$ increased the thyroid $V_{40}$ ($P < 0.05$). MU reductions averaged 5.1% ($S_{50}$), 21.4% ($S_{80}$), and 30.9% ($S_{100}$),

**Data availability statement:** All relevant data are within the paper and its Supporting Information files.

**Funding:** The author(s) received no specific funding for this work.

**Competing interests:** The authors have declared that no competing interests exist.

with consistent beam delivery times (~2.5 minutes). Gamma passing rates improved sequentially from the base plan to $S_{50}$, $S_{80}$, and $S_{100}$.

## Conclusion

MU optimization in NPC VMAT planning effectively reduces MUs and enhances delivery accuracy (improved gamma passing rates). While target coverage and OAR sparing were generally maintained, higher MUO strengths (e.g., $S_{100}$) may necessitate careful consideration of dosimetric trade-offs. Moderate MUO settings (e.g., $S_{80}$) offer a favorable balance between MU reduction and plan fidelity.

## Introduction

Nasopharyngeal carcinoma (NPC) is one of the most common malignant cancers in China, particularly in the southeastern regions, where the incidence rate ranges from 20 to 50 per 100,000 individuals [1]. Given the anatomical complexity of the nasopharynx and the tumor's high radiosensitivity, radiotherapy remains the primary treatment modality [2]. Radiotherapy techniques have evolved from three-dimensional conformal radiotherapy (3DCRT) to fixed-field intensity-modulated radiotherapy (IMRT), and more recently, to volumetric modulated arc therapy (VMAT) [3]. These advancements have significantly improved the clinical outcomes for NPC patients [4,5]. Notably, VMAT offers superior normal tissue sparing, reduces early adverse effects, and enhances survival rates compared to IMRT [6]. Additionally, VMAT achieves excellent target coverage while minimizing doses to organs at risk (OARs), particularly neural structures and salivary glands [7,8].

VMAT delivers radiation through continuous adjustments of gantry rotation speed, dose rate, and multi-leaf collimator (MLC) movement during treatment [9]. Compared to IMRT, VMAT significantly reduces monitor units (MUs), shortens treatment duration, and improves delivery efficiency [10,11]. Lower MUs also decrease the risk of secondary malignancies induced by scattered radiation [12].

The Eclipse treatment planning system (TPS) provides multiple optimization tools, including generalized equivalent uniform dose optimization, dose-volume histogram (DVH)-based structure optimization, normal tissue objective optimization, and the monitor unit objective (MUO) tool [13–16]. The MUO tool regulates MUs in VMAT plan through three adjustable parameters: *Minimum MU, Maximum MU, and Strength*. *Minimum MU* and *Maximum MU* define the lower and upper bounds for MUs, respectively, while *Strength* determines the priority of MU reduction (Fig 1). Previous studies have applied the MUO tool in VMAT plan for hypopharyngeal, prostate, cervical, breast, and non-small cell lung cancers [17–20]. These studies demonstrated that judicious MUO use reduces total MUs without compromising target coverage or OAR doses. However, research on MUO optimization in NPC-specific VMAT plans remains limited. NPC poses unique challenges due to its intricate anatomy, high target dose requirements, and close proximity to multiple OARs [21]. This study investigates the impact of MUO-driven optimization on VMAT plan quality for NPC.

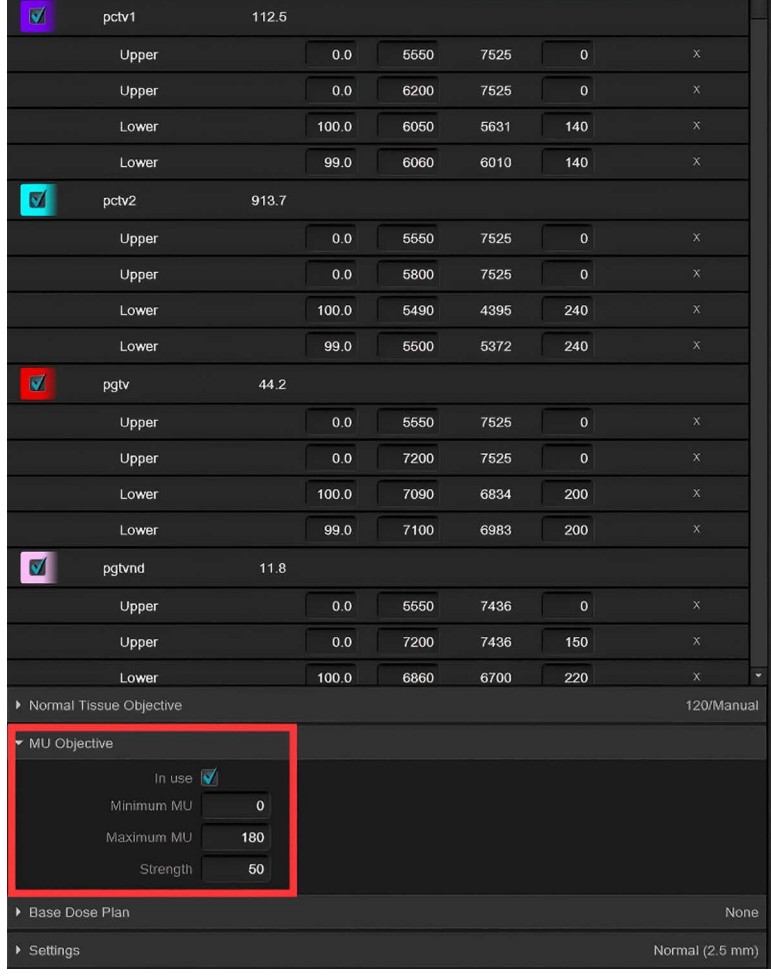

**Fig 1. The optimization function interface for VMAT plan in the Eclipse TPS.** The settings include dose-volume objectives, normal tissue objectives, and MU objective, where MU objective are used to control the total MU of the VMAT plan.

## Materials and methods

### Patient cohort

This retrospective study received approval from the Institutional Review Board at Gaozhou People's Hospital (GYLLPJ-2023027). We enrolled 21 patients with NPC treated with VMAT from October 1, 2020 to March 1, 2021. The requirement for informed consent was waived as this is a retrospective study for research purposes only. Data collection and analysis occurred from October and December 2024. All patient data were anonymized prior to analysis. The cohort comprised 18 males and 3 females, with ages ranging from 31 to 76 years (median: 45 years; mean±SD: 51±12 years). Inclusion criteria were: (1) histopathological confirmation of NPC diagnosis, and (2) completion of the prescribed radiotherapy course. We excluded patients who: (1) lacked complete pretreatment imaging or pathological records, or (2) presented with cachexia at diagnosis.

### CT simulation and target delineation

Patients were immobilized in a thermoplastic head-neck-shoulder mask (Orfit Industries, Wijnegem, Belgium) with a head rest, maintaining a standardized supine position with arms positioned alongside the body. Simulation CT scans were

acquired using a Siemens SOMATOM Confidence large-bore CT simulator (Siemens Healthineers, Forchheim, Germany), with the following scanning settings: scan range extending from the cranial vertex to 5 cm inferior to the aortic arch, 3 mm slice thickness. All CT datasets were then transferred to the Eclipse TPS (version 13.6; Varian Medical Systems, Palo Alto, CA) for target volume and OAR delineation. Target volumes were defined as follows: Gross tumor volume (GTV): Primary tumor delineated based on fused diagnostic MRI and clinical findings, GTVnd: Bilateral metastatic lymph node, CTV1: High-risk subclinical disease volume, CTV2: Low-risk subclinical disease volume. Each target volume received a uniform 3 mm isotropic expansion to generate the corresponding planning target volumes (PGTV, PGTVnd, PCTV1, and PCTV2). OARs were contoured according to established guidelines and included: Central nervous system structures (spinal cord, brainstem, optic chiasm, bilateral optic nerves), Sensory organs (bilateral eyes, lenses), Endocrine structures (pituitary gland), Salivary glands (bilateral parotid glands), Neurological and musculoskeletal structures (temporal lobes, temporo-mandibular joints).

### Dose prescription

Treatment plans employed a simultaneous integrated boost technique with the following prescription doses delivered in 32 fractions (5 fractions weekly): 7000 cGy for PGTV, 6800 cGy for PGTVnd, 6000 cGy for PCTV1, and 5400 cGy for PCTV2. Dose coverage objectives required that ≥95% of each target volume receive 100% of the prescribed dose. OAR constraints were implemented as follows: brainstem ≤ 5400 cGy, spinal cord ≤ 4500 cGy, optic chiasm and nerves ≤ 5400 cGy, lenses ≤ 1000 cGy, eyes ≤ 4500 cGy, For the parotid glands, the mean dose to at least one gland was constrained to ≤ 2600 cGy, with $D_{50\%}$ < 3000 cGy. The temporomandibular joint was limited to < 6000 cGy. All target volume definitions and plan evaluations adhered to the Radiation Therapy Oncology Group (RTOG) 0615 protocol [22] and the International Commission on Radiation Units and Measurements (ICRU) Report 83 [23].

### VMAT planning

For each patient, we implemented a dual-arc VMAT delivery strategy with the following specifications. The first arc employed counterclockwise gantry rotation from 179° to 181° with a collimator angle of 10°, while the second arc used clockwise rotation from 181° to 179° with a collimator angle of 350°. All plans utilized a 6 MV photon beam with a maximum dose rate of 600 MU/min and were optimized using a photon optimization algorithm in Varian Eclipse v13.6 TPS. The initial optimization created a "base" plan without MUO tool. Subsequently, we generated three additional plans by re-optimizing only the MUO parameters while maintaining identical field geometry, dose-volume objectives, and structure weighting factors. Based on Prado et al.'s findings [24], we set the Minimum MU to 0 and Maximum MU to 30% of the base plan's total MUs, while testing Strength parameters of 50, 80, and 100. This approach yielded four distinct plans per patient: (1) the original "base" plan without MUO; (2) "$S_{50}$" with Strength = 50; (3) "$S_{80}$" with Strength = 80; and (4) "$S_{100}$" with Strength = 100. The 30% maximum MU constraint was specifically chosen based on previous evidence demonstrating its effectiveness in MU reduction while preserving plan quality [24]. This systematic evaluation of Strength parameters allowed comprehensive assessment of the MUO tool's sensitivity in NPC VMAT planning.

### Comparison of target and OAR doses, MUs, and beam delivery time

We conducted a comprehensive comparison between the three MUO-optimized plans ($S_{50}$, $S_{80}$, and $S_{100}$) and the base plan using multiple evaluation parameters. For target volumes, we analyzed the $D_{95\%}$ coverage for all planning target volumes (PGTV, PGTVnd, PCTV1, and PCTV2), along with maximum dose ($D_{max}$), mean dose ($D_{mean}$), and homogeneity index (HI) as defined in ICRU Report 83 [23], OAR evaluation included $D_{max}$ assessment for critical serial organs (brainstem, spinal cord, optic chiasm, optic nerves, lenses, eyes, and pituitary gland), $D_{mean}$ for parallel organs (parotid glands, thyroid gland, and submandibular glands), and specific dose-volume parameters including $D_{50\%}$ for parotid glands and $V_{40}$

(volume receiving ≥40 Gy) for thyroid gland. Treatment efficiency was quantified by monitoring the total MUs and precisely measuring beam delivery time using a calibrated stopwatch. The delivery time measurement specifically captured the beam-on to beam-off duration for each arc, deliberately excluding setup time and beam mode up time, with the total delivery time representing the sum of both arcs' active treatment durations. This rigorous evaluation protocol enabled systematic comparison of plan quality versus delivery efficiency across all optimization approaches while maintaining consistent measurement standards.

### Pretreatment plan verification

For pre-treatment plan quality verification, the 3D dose distributions of all four VMAT plans (base, $S_{50}$, $S_{80}$, and $S_{100}$) were measured for each patient using the Sun Nuclear ArcCheck diode array phantom (Sun Nuclear Inc, Melbourne, FL) mounted on a Varian Trilogy linear accelerator (Varian Medical Systems, Palo Alto, CA). The measured dose distributions were compared with the TPS-calculated doses using gamma index analysis in SNC Patient software (Sun Nuclear Inc, Melbourne, FL). The primary evaluation used a global absolute dose gamma evaluation of a 3%/3 mm with a 10% dose threshold [25]. Prior to measurements, the output dose of the Trilogy linear accelerator was calibrated, following the International Atomic Energy Agency TRS-398 protocol [26], to ensure dosimetric accuracy. To further assess the impact of MU reduction on plan delivery accuracy, additional gamma analyses were performed under stricter criteria: 2%/2 mm, 3%/2 mm, and 1%/1 mm. This multi-criteria evaluation allowed systematic investigation of whether reduced MUs influenced the gamma passing rate (GPR) while maintaining clinically acceptable plan quality.

### Statistical analysis

All statistical analysis was performed using SPSS Statistics version 17.0 (IBM Corp., Armonk, NY, USA). We first assessed data normality using the Shapiro-Wilk test. Normally distributed data were presented as mean ± standard deviation ($\bar{x} \pm s$). while non-normally distributed data were expressed as median (interquartile range). For normally distributed parameters, paired Student's t-tests were conducted to compare the MUO-optimized plans ($S_{50}$, $S_{80}$, and $S_{100}$) with the base plan, otherwise, the multi-correlation sample non-parametric Friedman test is used. The significance level was set at $P < 0.05$ (two-tailed).

## Results

### Target dose comparison

The target dose parameters across the four treatment plans (base, $S_{50}$, $S_{80}$, and $S_{100}$) were compared, with detailed results presented in Tables 1 and 2. Figs 2–4 illustrate the relative differences in target dose parameters between the MUO-optimized plans and the base plan for all 21 patients.

 Plan $S_{50}$ showed statistically significant differences ($P < 0.05$) in several key parameters compared to the base plan: the $D_{mean}$ of PGTV, $D_{95\%}$ and HI of PGTVnd, and $D_{mean}$ and HI of PCTV2. Importantly, all other dose parameters maintained comparable values ($P > 0.05$), demonstrating that the $S_{50}$ optimization preserved most dosimetric characteristics while achieving MU reduction.

 More extensive variations emerged in Plan $S_{80}$, with significant differences ($P < 0.05$) observed in multiple parameters: $D_{95\%}$ and HI of PGTV; $D_{95\%}$, $D_{mean}$, and HI of PGTVnd; and $D_{95\%}$, $D_{max}$, and HI of PCTV2. However, other dose parameters remained unchanged, suggesting a selective impact of the increased MUO strength.

 The most pronounced differences occurred in Plan $S_{100}$, where only PGTV $D_{max}$ and $D_{mean}$, PGTVnd $D_{max}$, and PCTV1 $D_{max}$ showed no significant variation ($P > 0.05$). Clinically relevant dosimetric changes included a > 4% decrease in PCTV2 $D_{98\%}$ for 4 patients (19%) and 3–4% reductions in PGTV and PGTVnd $D_{max}$ and PCTV2 $D_{98\%}$ for 1 patient (5%). Most other parameter changed within 2%.

**Table 1. Dose comparisons of target PGTV and PGTV$_{nd}$ for plan S$_{50}$, S$_{80}$, S$_{100}$ vs base plan ($n=21$).**

| plan | PGTV | | | | PGTV$_{nd}$ | | | |
|---|---|---|---|---|---|---|---|---|
| | $D_{95\%}$/cGy | $D_{max}$/cGy | $D_{mean}$/cGy | HI | $D_{95\%}$/cGy | $D_{max}$/cGy | $D_{mean}$/cGy | HI |
| **base** | 7 080.0±13.0 | 7 550.3±68.8 | 7 230.4±11.7 | 0.047±0.004 | 6 892.3±19.2 | 7 455.5±135.7 | 7 082.7±32.5 | 0.061±0.007 |
| **S$_{50}$** | 7 074.4±15.56 | 7 548.0±66.2 | 7 218.3±8.9 | 0.046±0.005 | 6 882.0±23.9 | 7 441.0±122.4 | 7 079.0±35.9 | 0.063±0.008 |
| **S$_{80}$** | 7 069.9±18.6 | 7 531.4±67.7 | 7 230.8±14.1 | 0.050±0.005 | 6 857.2±24.7 | 7 429.6±134.6 | 7 065.8±34.8 | 0.068±0.012 |
| **S$_{100}$** | 7 050.4±28.0 | 7 231.9±14.5 | 7 231.9±14.5 | 0.055±0.008 | 6 822.4±51.6 | 7 443.3±107.3 | 7 041.4±44.2 | 0.075±0.014 |
| **P$_1$** | 0.076 | 0.800 | 0.002 | 0.205 | 0.008 | 0.285 | 0.281 | 0.012 |
| **P$_2$** | 0.008 | 0.182 | 0.912 | 0.002 | <0.001 | 0.109 | <0.001 | 0.001 |
| **P$_3$** | <0.001 | 0.088 | 0.683 | <0.001 | <0.001 | 0.409 | <0.001 | <0.001 |

Note: $P_1$ indicates the test results of plan S$_{50}$ and base plan; $P_2$ indicates the test results of plan S$_{80}$ and base plan. $P_3$ indicates the test result of plan S$_{100}$ and base plan.

**Table 2. Dose comparisons of target PCTV1 and PCTV2 for plan S$_{50}$, S$_{80}$, S$_{100}$ vs base plan ($n=21$).**

| plan | PCTV1 | | | | PCTV2 | | | |
|---|---|---|---|---|---|---|---|---|
| | $D_{95\%}$/cGy | $D_{max}$/cGy | $D_{mean}$/cGy | HI | $D_{95\%}$/cGy | $D_{max}$/cGy | $D_{mean}$/cGy | HI |
| **base** | 6 261.4±105.4 | 7 556.3±74.0 | 6 915.8±83.8 | 0.174±0.014 | 5 513.6±23.4 | 7 592.8±88.5 | 6 236.2±69.6 | 0.311±0.005 |
| **S$_{50}$** | 6 269.7±98.4 | 7 548.8±66.6 | 6 915.5±80.6 | 0.172±0.012 | 5 520.7±26.9 | 7 576.8±75.2 | 6 248.3±71.7 | 0.307±0.006 |
| **S$_{80}$** | 6 249.9±106.7 | 7 531.9±67.3 | 6 913.9±82.2 | 0.176±0.015 | 5 479.3±38.6 | 7 555.1±92.9 | 6 233.4±72.6 | 0.318±0.010 |
| **S$_{100}$** | 6 214.4±114.0 | 7 535.0±68.4 | 6 902.9±83.9 | 0.184±0.017 | 5 417.5±66.2 | 7 564.3±68.2 | 6 198.0±81.5 | 0.334±0.018 |
| **P$_1$** | 0.235 | 0.465 | 0.882 | 0.139 | 0.127 | 0.170 | <0.001 | <0.001 |
| **P$_2$** | 0.145 | 0.127 | 0.481 | 0.060 | <0.001 | 0.033 | 0.462 | 0.002 |
| **P$_3$** | <0.001 | 0.081 | <0.001 | <0.001 | <0.001 | 0.047 | <0.001 | <0.001 |

Note: $P_1$ indicates the test results of plan S$_{50}$ and base plan; $P_2$ indicates the test results of plan S$_{80}$ and base plan. $P_3$ indicates the test result of plan S$_{100}$ and base plan.

Visual analysis of Fig 5 (cross-sectional dose distribution) revealed progressive extension of the 40 Gy isodose line toward the mandible across S$_{50}$→S$_{80}$→S$_{100}$ plans, while the DVH in Fig 6 provided dose-volume relationships. These findings demonstrate a clear trade-off between MU reduction efficacy and target coverage preservation, with S$_{80}$ appearing to represent an optimal balance between these competing objectives. The progressive changes across the plans suggest that MUO strength requires careful selection to maintain clinical dosimetric goals while achieving efficiency gains.

## OARs dose comparison

The comparative assessment of OAR doses across the four treatment plans revealed several important findings (Tables 3,4). All MUO-optimized plans (S$_{50}$, S$_{80}$, S$_{100}$) demonstrated significant increases ($P<0.05$) in brainstem D$_{max}$ compared to the base plan, though the absolute differences remained clinically modest (<1.5%). Notably, plan S$_{100}$ showed additional elevation in spinal cord D$_{max}$ ($P<0.05$).

Parotid gland analysis revealed distinct optimization patterns: plan S$_{50}$ achieved reductions in both D$_{50\%}$ and D$_{mean}$ for bilateral parotids ($P<0.001$), while plan S$_{80}$ maintained D$_{mean}$ reduction bilaterally ($P<0.001$) but only reduced left parotid D$_{50\%}$ ($P<0.001$ vs $P=0.181$ for right). In contrast, plan S$_{100}$ increased both dose parameters for parotids ($P<0.05$). Thyroid gland evaluation showed V$_{40}$ increases in both S$_{80}$ and S$_{100}$ plans ($P<0.05$).

Critical structures including bilateral temporal lobes (D$_{max}$), temporomandibular joints (D$_{max}$), and submandibular glands (D$_{mean}$) maintained stable dose distributions across all plans without significant variations ($P>0.05$). These findings

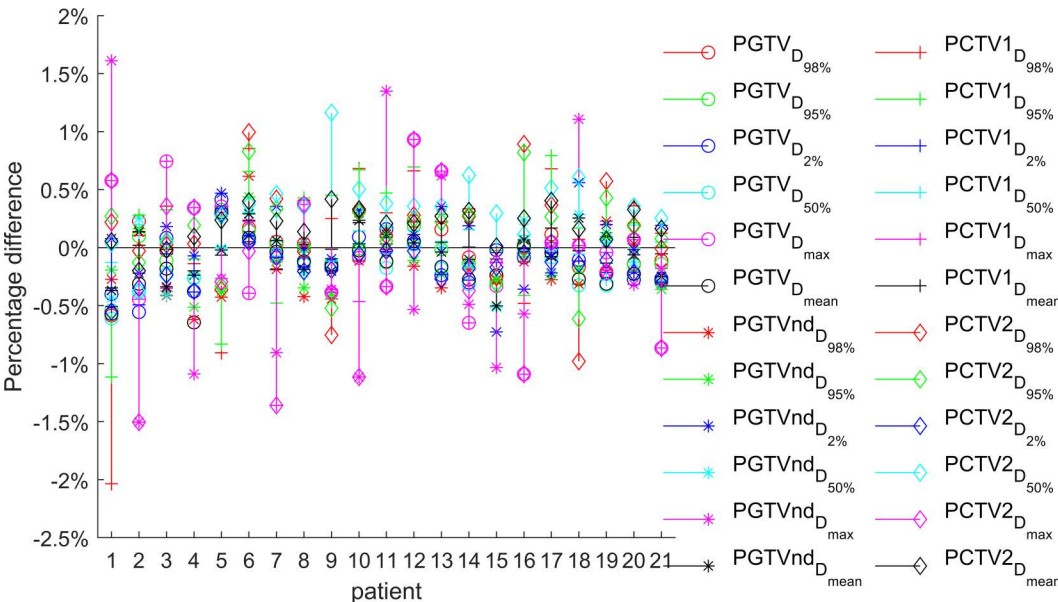

**Fig 2. The relative difference between the target dose of plan S$_{50}$ and base plan (n =21).** Among the 21 patients in plan S$_{50}$, only one showed a decrease in target dose exceeding 2% but remaining below 2.5% compared to the base plan. For the PGTV, dose deviations were within 1% of the base plan.

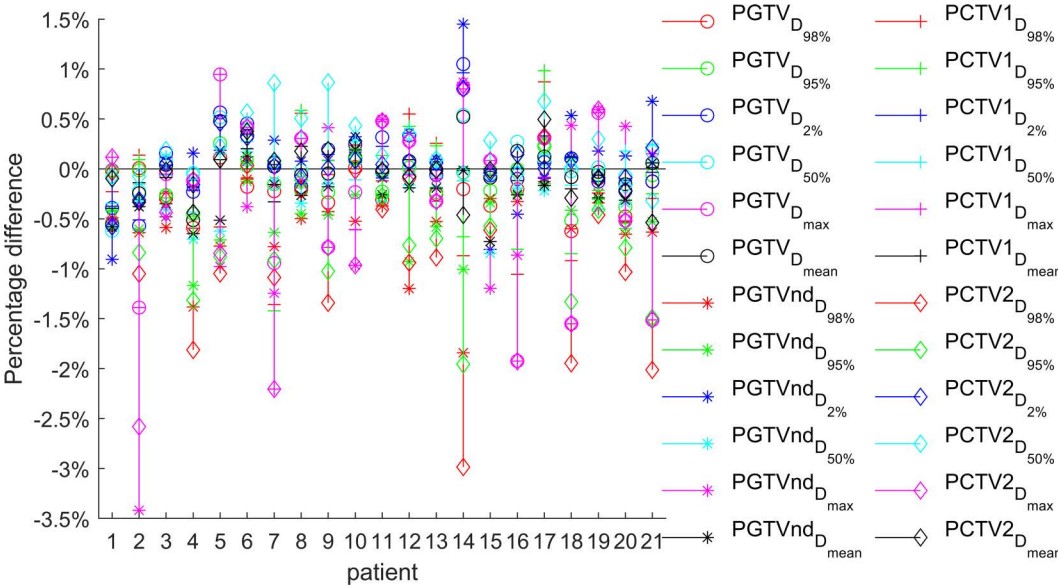

**Fig 3. The relative difference between the target dose of plan S$_{80}$ and base plan (n =21).** Among the 21 patients in Plan S$_{80}$, only one case showed a deviation of D$_{95\%}$ exceeding 1.5% but remaining below 2% compared to the base plan. For PGTV, all cases D$_{95\%}$ deviations were within 1% of the base plan.

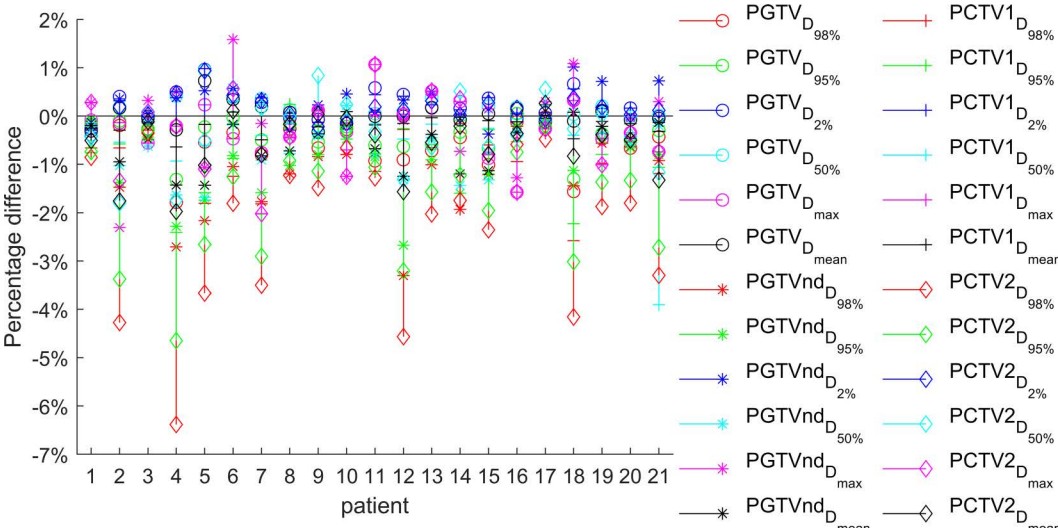

**Fig 4. The relative difference between the target dose of plan $S_{100}$ and base plan (n = 21).** In Plan $S_{100}$, the PCTV2 showed a notable dose reduction compared to the base plan. Among the cases, four patients demonstrated a decrease in $D_{95\%}$ exceeding 3%, while another four patients had a $D_{98\%}$ reduction surpassing 4%. However, for the PGTV, all dose deviations from the base plan were within 2%.

demonstrate that while MUO optimization affects certain OARs differentially based on strength setting, it preserves dose constraints to several critical neural and musculoskeletal structures.

The results suggest a clear trade-off between MU reduction efficacy and OAR sparing, with plan $S_{50}$ representing the most favorable balance – achieving significant parotid dose reduction without compromising other critical structures. Higher strength optimizations (particularly $S_{100}$) showed diminishing returns, introducing dose elevations to multiple OARs while providing only marginal additional MU reduction benefits. These findings have important clinical implications for NPC radiotherapy planning, where parotid sparing is particularly crucial for maintaining patient quality of life.

**MU and beam delivery time comparison**

The analysis of treatment delivery efficiency revealed important findings (Table 5). Compared to the base plan, all MUO-optimized plans demonstrated statistically significant reductions in MUs ($P < 0.05$). Plan $S_{50}$ achieved a modest 5.1% average MU reduction, while $S_{80}$ and $S_{100}$ showed more substantial decreases of 21.5% and 30.9%, respectively. This progressive reduction pattern correlated directly with increasing MUO strength parameters.

Interestingly, despite these significant MU reductions, the beam-on time remained consistent across all plans. The average delivery time maintained approximately 2.5 minutes for all four treatment plans (base, $S_{50}$, $S_{80}$, and $S_{100}$). This observation suggests that while the MUO tool effectively reduces total MUs - potentially lowering secondary cancer risks from scattered radiation – it does not translate into faster treatment delivery under current operational parameters.

**Gamma passing rate comparison**

The GPR evaluation of the four treatment plans under different gamma criteria (3%/3 mm, 2%/2 mm, 3%/2 mm, and 1%/1 mm) revealed important insights into plan delivery accuracy (Table 6). Plan $S_{50}$, with a modest 5.1% MU reduction, demonstrated equivalent GPR performance to the base plan ($P > 0.05$), indicating that limited MU optimization maintains delivery accuracy. In contrast, both Plan $S_{80}$ (21.5% MU reduction) and Plan $S_{100}$ (30.9% MU reduction) showed differences in GPR compared to the base plan ($P < 0.001$) under all evaluation conditions.

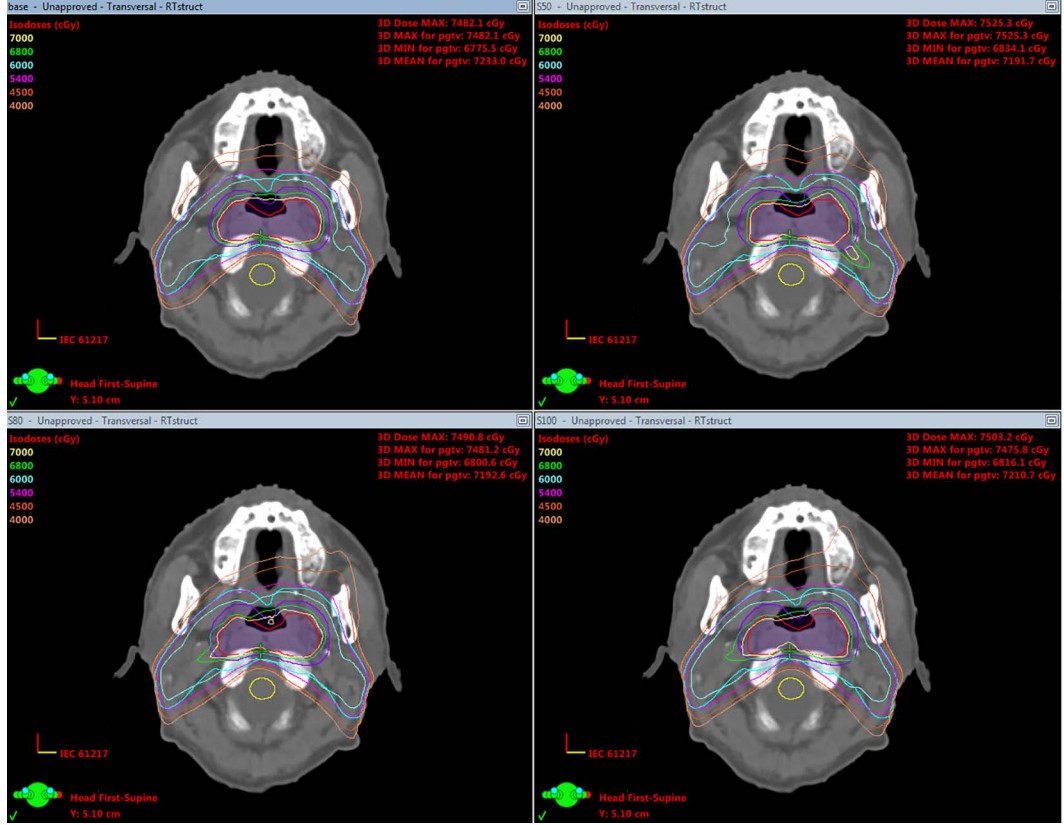

**Fig 5. Cross-sectional dose distribution of one patient, with the top left, top right, bottom left, and bottom right indicating planned base, S$_{50}$, S$_{80}$, and S$_{100}$, respectively.** The 7000 cGy isodose lines (yellow) in all four plans demonstrated adequate coverage of the target PGTV (red). However, in Plan S$_{100}$, the 7000 cGy isodose line showed inward retraction and failed to completely cover the PGTV. Compared with the base plan, Plan S$_{50}$, S$_{80}$ and S$_{100}$ exhibited progressively expanding low-dose isodose lines (4500 cGy and 4000 cGy), resulting in deteriorated dose conformity.

A particularly interesting finding emerged when examining the relationship between MU reduction and delivery precision. Under the most stringent 1%/1mm criterion, we observed a consistent trend where greater MU reduction correlated with improved GPR. This relationship suggests that the reduced MU variability in optimized plans may enhance delivery precision, particularly for more aggressive optimization settings. However, this potential benefit must be carefully weighed against the previously noted trade-offs in target coverage and OAR sparing observed with higher MUO strength settings.

## Discussion

Modern radiotherapy has revolutionized the treatment of NPC, with IMRT establishing itself as the preferred treatment method by achieving 5-year local control rates of approximately 90% [27]. The subsequent development of VMAT has further advanced the field, offering significant improvements over conventional IMRT through its dynamic delivery system. VMAT's arc-based optimization provides superior dose conformity to complex tumor geometries while simultaneously enhancing protection of surrounding critical structures [28]. These technical advantages have translated into tangible clinical benefits, most notably in reducing the incidence and severity of radiation-induced late complications that impact patients' quality of life. Clinical studies have consistently demonstrated VMAT's ability to mitigate treatment-related toxicities, including sensorineural hearing loss, trismus, and temporal lobe injury [29–31]. The technique's improved efficiency,

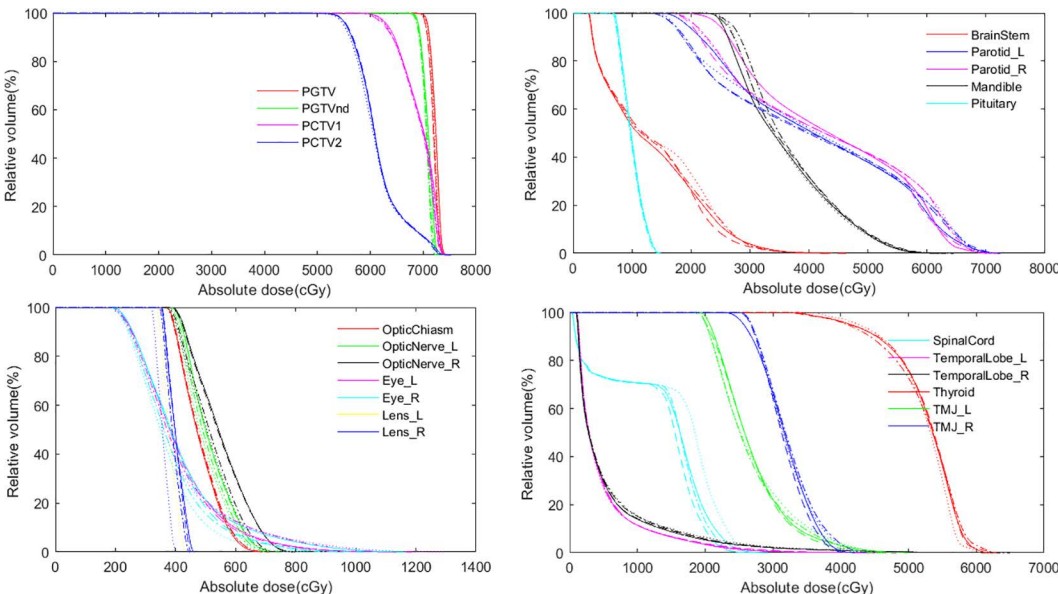

**Fig 6. DVH of each structure of four plans for one patient, where "-solid line" represents base plan, "-- long dashed line" represents plan S$_{50}$, "-. dotted dash line" represents plan S$_{80}$, and ": dotted line" represents plan S$_{100}$.** For the target, plan S$_{100}$ showed significant deviations in the DVH compared to the base plan. Regarding OARs, plan S$_{100}$ also exhibited notable DVH variations for the parotid glands and spinal cord. In contrast, plan S$_{50}$ demonstrated nearly identical DVH curves to the base plan for both target and OARs.

**Table 3. Dose comparison of brainstem, lens, chiasma, optic nerve, pituitary gland and spinal cord for plan S$_{50}$, S$_{80}$, S$_{100}$ vs base plan (_n_ = 21).**

| plan | brainstem $D_{max}$/ cGy | left lens $D_{max}$/cGy | right lens $D_{max}$/cGy | optic chiasma $D_{max}$/cGy | left optic nerve $D_{max}$/cGy | right optic nerve $D_{max}$/cGy | pituitary $D_{max}$/cGy | spinal cord $D_{max}$/cGy |
|---|---|---|---|---|---|---|---|---|
| base | 5 512.1±648.7 | 682.5±187.9 | 668.9±156.5 | 3 807.8±2 359.4 | 3 394.8±2 152.8 | 4 215.2±1 820.8 | 5 093.6±1 825.0 | 2 873.4±299.1 |
| S$_{50}$ | 5 566.8±615.8 | 681.0±171.0 | 676.5±162.0 | 3 798.9±2 398.3 | 3 331.6±2 188.9 | 4 230.5±1 861.1 | 5 109.0±1 834.6 | 2 755.0±356.4 |
| S$_{80}$ | 5 562.5±619.1 | 681.3±180.7 | 666.2±185.0 | 3 775.9±2 401.2 | 3 366.6±2 217.8 | 4 228.7±1 856.0 | 5 112.6±1 854.5 | 2 920.3±369.9 |
| S$_{100}$ | 5 590.1±631.0 | 693.8±227.6 | 670.2±213.4 | 3 787.0±2 418.7 | 3 338.4±2 219.1 | 4 220.4±1 860.6 | 5 128.9±1 852.4 | 3 317.2±433.7 |
| $P_1$ | 0.025 | 0.839 | 0.231 | 0.644 | 0.029 | 0.455 | 0.605 | 0.016 |
| $P_2$ | 0.024 | 0.865 | 0.850 | 0.337 | 0.422 | 0.392 | 0.650 | 0.334 |
| $P_3$ | 0.013 | 0.396 | 0.944 | 0.472 | 0.208 | 0.820 | 0.404 | <0.001 |

Note: $P_1$ indicates the test results of plan S$_{50}$ and base plan; $P_2$ indicates the test results of plan S$_{80}$ and base plan. $P_3$ indicates the test result of plan S$_{100}$ and base plan.

characterized by shorter treatment delivery times, further enhances its clinical utility without compromising therapeutic outcomes.

The MUO tool has been investigated across various treatment sites with varying results. While Varian established the basic functionality of the MUO tool, clinical implementation strategies have primarily been developed through subsequent research. Jiménez-Puertas et al. demonstrated that setting Maximum MU to 400 could achieve 15–22% MU reduction in prostate, gynecological, and head-neck VMAT plans without compromising dosimetric quality, though their study did

**Table 4. Dose comparison of parotid gland, temporal lobe, temporomandibular joint, thyroid gland and submandibular gland for plan $S_{50}$, $S_{80}$, $S_{100}$ vs base plan (n = 21).**

| Structure | Parameters | base | $S_{50}$ | $S_{80}$ | $S_{100}$ | $P_1$ | $P_2$ | $P_3$ |
|---|---|---|---|---|---|---|---|---|
| **Left parotid gland** | $D_{50\%}$/cGy | 3 228.0±443.8 | 3 094.4±520.0 | 3 145.3±480.2 | 3 431.3±440.8 | <0.001 | 0.001 | <0.001 |
| | $D_{mean}$/ cGy | 3 649.4±310.6 | 3 549.9±321.7 | 3 591.3±310.5 | 3 767.4±310.4 | <0.001 | <0.001 | <0.001 |
| right parotid gland | $D_{50\%}$/cGy | 3 551.8±618.2 | 3 439.1±728.7 | 3 505.5±686.4 | 3 684.8±624.4 | 0.004 | 0.181 | 0.003 |
| | $D_{mean}$/ cGy | 3 873.4±426.7 | 3 762.9±451.8 | 3 811.2±454.0 | 3 962.7±425.4 | <0.001 | <0.001 | 0.005 |
| left temporal lobe | $D_{max}$/cGy | 6 835.4±820.8 | 6 880.3±821.9 | 6 859.8±847.6 | 6 808.1±841.4 | 0.134 | 0.117 | 0.058 |
| right temporal lobe | $D_{max}$/cGy | 6 970.2±373.9 | 7 003.0±354.2 | 6 983.0±337.3 | 6 949.9±367.6 | 0.273 | 0.602 | 0.467 |
| left temporomandibular joint | $D_{max}$/cGy | 5 235.8±939.2 | 5 204.5±962.7 | 5 237.4±899.6 | 5 224.1±817.2 | 0.494 | 0.971 | 0.810 |
| right temporomandibular joint | $D_{max}$/cGy | 4 970.6±685.4 | 4 963.0±769.5 | 4 972.2±750.0 | 4 992.9±731.8 | 0.908 | 0.973 | 0.706 |
| thyroid gland | $D_{mean}$/cGy | 4 955.2±225.1 | 5 017.6±335.5 | 4 787.4±1 005.0 | 5 108.3±287.0 | 0.286 | 0.414 | 0.006 |
| | $V_{40}$/% | 87.2±7.9 | 88.0±7.4 | 90.6±6.5 | 94.5±5.0 | 0.312 | 0.002 | <0.001 |
| submandibular gland | $D_{mean}$/cGy | 5 779.7±384.4 | 5 800.7±391.4 | 5 751.4±390.9 | 5 737.8±422.9 | 0.066 | 0.059 | 0.062 |

Note: $P_1$ indicates the test results of plan $S_{50}$ and base plan; $P_2$ indicates the test results of plan $S_{80}$ and base plan. $P_3$ indicates the test result of plan $S_{100}$ and base plan.

**Table 5. MUs and beam on time for plan $S_{50}$, $S_{80}$, $S_{100}$ vs base plan (n = 21).s.**

| | MUs | Time(min) |
|---|---|---|
| base | 670±84.7 | 2.50±0.05 |
| $S_{50}$ | 636±57.9 | 2.50±0.06 |
| $S_{80}$ | 526±61.9 | 2.48±0.05 |
| $S_{100}$ | 463±65.0 | 2.48±0.05 |
| $t_1$ | 3.239 | 0.977 |
| $P_1$ | 0.004 | 0.340 |
| $t_2$ | 9.652 | −0.568 |
| $P_2$ | <0.001 | 0.576 |
| $t_3$ | 10.403 | −0.204 |
| $P_3$ | <0.001 | 0.841 |

Note: $P_1$ indicates the test results of plan $S_{50}$ and base plan; $P_2$ indicates the test results of plan $S_{80}$ and base plan. $P_3$ indicates the test result of plan $S_{100}$ and base plan. $t_1$, $t_2$, and $t_3$ represent the t-values from paired t-tests comparing plan $S_{50}$, $S_{80}$, and $S_{100}$ with the base plan, respectively.

not examine the impact of the Strength parameter [15]. Clemente et al.'s prostate cancer study revealed an important dose-quality trade-off, with Strength = 100 achieving 28% MU reduction but worsening target homogeneity by 23% [18]. Huang et al. applied the MUO tool to non-small cell lung cancer stereotactic body radiation therapy plans, demonstrating successful MU reduction while maintaining OAR doses [19]. Mancosu et al. investigated MUO tool for VMAT breast treatments. They fixed Strength = 100 and Maximum MU used were −50%, −20%, 20% and 50% of the reference MU plans. They found no significant deviations in target PTV coverage while for OARs some unacceptable deviations were observed. The MU reduction obtained range from 18% to 39% [20]. Our findings in NPC show similar trends, with Strength = 100 producing 31% MU reduction but 17% homogeneity degradation in PGTV, suggesting this represents a characteristic effect of high-strength MUO optimization.

The MUO-optimized plans achieved progressively greater MU reductions from $S_{50}$ (5.1%) to $S_{100}$ (30.9%), all statistically significant (P < 0.004). However, this efficiency came at variable clinical costs. Most notably, the highest-efficiency

**Table 6. Gamma pass rates of the four plans under different evaluation conditions.**

| plan | 3%/3 mm | 2%/2 mm | 3%/2 mm | 1%/1 mm |
|------|---------|---------|---------|---------|
| base | 99.7%±0.3% | 98.3%±1.0% | 99.4%±0.5% | 86.7%±4.2% |
| $S_{50}$ | 99.6%±0.3% | 98.4%±1.2% | 99.4±0.4% | 87.1%±4.4% |
| $S_{80}$ | 99.8%±0.2% | 99.2%±0.8% | 99.7%±0.3% | 90.4%±4.7% |
| $S_{100}$ | 99.8%±0.2% | 99.3%±1.0% | 99.7%±0.4% | 92.3%±5.1% |
| $t_1$ | 0.188 | −0.401 | 0.167 | −0.653 |
| $P_1$ | 0.853 | 0.693 | 0.869 | 0.522 |
| $t_2$ | −3.414 | −6.598 | −4.077 | −6.327 |
| $P_2$ | 0.003 | <0.001 | 0.001 | <0.001 |
| $t_3$ | −3.111 | −5.678 | −3.313 | −7.191 |
| $P_3$ | 0.006 | <0.001 | 0.004 | <0.001 |

Note: $P_1$ indicates the test results of plan $S_{50}$ and base plan; $P_2$ indicates the test results of plan $S_{80}$ and base plan. $P_3$ indicates the test result of plan $S_{100}$ and base plan. $t_1$, $t_2$, and $t_3$ represent the t-values from paired t-tests comparing plan $S_{50}$, $S_{80}$, and $S_{100}$ with the base plan, respectively.

$S_{100}$ plans showed clinically unacceptable target underdosing in 38% of cases (8/21), with PCTV2 $D_{98\%}$ deviations up to 6% that could potentially compromise tumor control. Intermediate $S_{80}$ plans maintained better dosimetric fidelity while still achieving 21.5% MU reduction, with most deviations remaining below clinically relevant thresholds (<1% or <50 cGy total dose). The minimal OAR dose increases (≤1.5% $D_{max}$ for critical neural structures) across all plans suggest these may represent inherent system characteristics rather than optimization failures.

These findings carry important clinical implementation recommendations. For centers prioritizing dose fidelity, conservative $S_{50}$ optimization provides modest MU reduction (5.1%) with minimal clinical impact. The $S_{80}$ setting offers a balanced compromise (21.5% MU reduction) suitable for most routine cases, while $S_{100}$ optimization (30.9% reduction) should be reserved for specific clinical scenarios with rigorous quality assurance protocols. The anatomical dependence of these outcomes – particularly the heightened sensitivity of complex head-and-neck target volumes compared to other sites like breast or prostate – underscores the need for site-specific optimization guidelines rather than universal parameter sets. Future development should focus on hybrid algorithms that combine MUO with other optimization strategies to break the current efficiency-fidelity trade-off paradigm.

Our study revealed important insights into how MU reduction impacts various aspects of treatment delivery and plan quality. Notably, the beam-on time remained consistent at approximately 2.5 minutes across all plans regardless of MU reduction level, which can be attributed to the Eclipse TPS photon optimization algorithm's tendency to maintain a relatively constant gantry rotation speed of 6.0°/s. This finding demonstrates that MU reduction primarily affects dose rate modulation and MLC motion speed rather than overall treatment duration.

We observed that MU reduction was associated with improved GPR, particularly under stringent 1%/1 mm evaluation criteria. The $S_{50}$, $S_{80}$, and $S_{100}$ plans showed progressive GPR improvements of 0.4%, 3.7%, and 5.6% compared to the base plan respectively. This result was contributed from several factors: (1) generation of simpler MLC shapes, (2) more consistent dose rate change patterns, and (3) reduced out-of-field scatter dose. These findings suggest that moderate MU reduction may actually enhance plan quality by creating more reproducible beam delivery sequences.

The clinical implications of these results are significant. For complex treatment plans that initially show marginal GPR (<95% under 3 mm/3% criteria), judicious application of the MUO tool could serve dual purposes – both reducing MUs and improving delivery accuracy. Additionally, the reduction in total MUs directly translates to decreased scatter dose to patients, potentially lowering the risk of secondary radiation-induced malignancies [32]. This benefit is particularly relevant for younger patients or those with long expected survival times.

This study has several important limitations that should be considered when interpreting the results. As a retrospective analysis, our investigation focused specifically on evaluating the dosimetric impacts of MUO optimization in NPC VMAT planning, including effects on target coverage, OAR doses, beam-on time, and GPR. However, we did not assess treatment endpoints such as tumor control probability or normal tissue complication probability, which represent important gaps in understanding the full clinical implications of our findings. Secondly, the advantages of using the MUO tool need to be confirmed through long-term patient follow-up and based on a large number of case practices.

## Conclusion

Our comprehensive evaluation of MUO-optimized VMAT plans for NPC provides clear guidance for clinical implementation. The study demonstrates that careful selection of MUO parameters can achieve significant MU reduction while maintaining treatment quality. Among the three optimization levels investigated, Plan $S_{80}$ emerges as the optimal choice for routine clinical practice, delivering a 21.5% MU reduction (P<0.001) while preserving target coverage and OAR protection. This moderate optimization level maintained all dosimetric parameters within clinically acceptable ranges and showed equivalent GPR to conventional plans, making it suitable for standard NPC treatment plans.

## Supporting information

**S1 File. Dose parameters for the target and OARs, as well as dose-volume histogram parameters for each plan of a representative patient.**
(ZIP)

## Author contributions

**Formal analysis:** Shukui Tang.

**Investigation:** Zongyou Chen.

**Methodology:** Zunbei Wen.

**Project administration:** Huaqu Zeng.

**Resources:** Qifu Lin.

**Software:** Zhen Li.

**Writing – original draft:** Minzhi Zhong.

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
