## [Decision Letter · Decision Letter 0]

PONE-D-25-12726Research on the Impact of Monitor Unit Optimization in Volumetric Modulated Arc Therapy Planning for Nasopharyngeal CarcinomaPLOS ONE

Dear Dr. zeng,

Thank you for submitting your manuscript to PLOS ONE. After careful consideration, we feel that it has merit but does not fully meet PLOS ONE’s publication criteria as it currently stands. Therefore, we invite you to submit a revised version of the manuscript that addresses the points raised during the review process.

We look forward to receiving your revised manuscript.

Kind regards,

Chung-Ta Chang

Academic Editor

PLOS ONE

Journal Requirements:

Reviewers' comments:

Reviewer's Responses to Questions

**Comments to the Author**

1. Is the manuscript technically sound, and do the data support the conclusions?

Reviewer #1: Yes

Reviewer #2: Yes

Reviewer #3: Yes

2. Has the statistical analysis been performed appropriately and rigorously? 

Reviewer #1: Yes

Reviewer #2: I Don't Know

Reviewer #3: Yes

3. Have the authors made all data underlying the findings in their manuscript fully available?

Reviewer #1: Yes

Reviewer #2: Yes

Reviewer #3: Yes

4. Is the manuscript presented in an intelligible fashion and written in standard English?

Reviewer #1: Yes

Reviewer #2: Yes

Reviewer #3: Yes

5. Review Comments to the Author

Reviewer #1: This manuscript presents a technically sound and clinically relevant analysis of monitor unit (MU) optimization in VMAT planning for nasopharyngeal carcinoma (NPC). The study leverages a well-described retrospective dataset of 21 patients and offers a thorough comparison between base plans and three optimization scenarios (S50, S80, and S100) using the Eclipse TPS's MUO tool.

The methodology is rigorous and clearly reported, with a reasonable range of Strength parameter values and fixed Maximum MU constraints. Dose coverage for target volumes, organ-at-risk sparing, MU reduction, beam-on time, and gamma passing rate (GPR) are assessed and interpreted appropriately. The data supports the conclusion that MU optimization—particularly with moderate Strength settings like S50 and S80—can meaningfully reduce total MUs without compromising clinical quality or delivery efficiency. The inclusion of multiple evaluation metrics and clear statistical treatment enhances the study's reliability.

Strengths:

-Clear clinical relevance: the findings directly support decision-making in VMAT plan optimization for NPC.

-Detailed dose comparisons for both target volumes and OARs.

-Inclusion of gamma index analysis under multiple criteria strengthens the dosimetric QA aspects.

-Ethical approval and data availability are appropriately addressed.

Areas for Improvement:

-Language and Style: While the manuscript is generally intelligible, some grammatical issues, tense inconsistencies, and awkward phrasing could be improved for clarity and flow. A light language edit by a native English speaker or language service would help.

-Discussion Depth: The discussion could benefit from a slightly more in-depth comparison with existing studies that evaluated Strength variation in MUO parameters—notably the clinical implications of underdosing in S100 could be expanded.

-Figures and Tables: The legends for Figures 1–5 could be made more detailed. Also, a summary figure or schematic explaining the MUO tool settings and their clinical impact would enhance readability.

Overall, the manuscript presents meaningful results that advance understanding of VMAT optimization in NPC and warrant publication after minor revisions.

Reviewer #2: The manuscript is nicely written by the author. some minor corrections I have mentioned on the original pdf as well as separate pdf file. As of my knowledge, MU reduction is achievable with this technique but many time, at the cost of target coverage (excluding S50).

Reviewer #3: This study investigates the dosimetric and delivery efficiency impacts of monitor unit (MU) optimization in VMAT for nasopharyngeal carcinoma (NPC). The topic is clinically relevant, as MU reduction can enhance treatment efficiency and potentially lower secondary cancer risks. The manuscript is well-structured, with clear methodology and robust statistical analysis. However, several areas require clarification and expansion to strengthen the scientific rigor and clinical applicability.

1.Clarify whether MUO was applied during initial optimization or as a post-hoc re-optimization. This affects reproducibility and clinical implementation.

2.Specify if OAR constraints were adjusted during MUO re-optimization. The observed increases in brainstem/spinal cord Dmax (≤1.5%) suggest constraints may need tightening.

3.Justify the choice of gamma criteria (1%/1 mm, 2%/2 mm, 3%/3 mm). While 1%/1 mm is stringent, its clinical relevance should be discussed (e.g., is 3%/2 mm more practical for routine QA?).

4.For OARs, discuss if the increases in brainstem Dmax (≤1.5%) or thyroid V40 are acceptable given clinical tolerances.

5.Highlight that S100’ MU reduction (30.9%) comes at the cost of target underdosing in 8/21 patients, making S50 or S80 more viable.

6.Explain why MU reduction did not shorten delivery time (e.g., fixed gantry speed, dose rate limitations).

7.Fixed MUO parameters (e.g., Maximum MU = 30% of base plan) may not be optimal for all cases. It is suggested to increase the discussion on the influence of the Maximum MU parameter on the results

8.Suggest prospective validation with larger cohorts, long-term toxicity outcomes, and exploration of adaptive MUO strategies.

9.In the Abstract, emphasize the improvement in gamma passing rates with MU reduction, as this is a notable practical outcome.

10.Ensure consistent formatting (e.g., italicize journal names, use "et al." correctly). The citation formats are inconsistent. Some authors use their full names while others use abbreviations.

6. PLOS authors have the option to publish the peer review history of their article (what does this mean? ). If published, this will include your full peer review and any attached files.

**Do you want your identity to be public for this peer review?** For information about this choice, including consent withdrawal, please see our Privacy Policy .

Reviewer #1: **Yes: ** Cosima C. Hoch

Reviewer #2: No

Reviewer #3: No

---

## [Author Response · Author response to Decision Letter 1]

3 Jun 2025

Dear Editors and Reviewers:

Thank you for your letter and for the reviewers’ comments concerning our manuscript entitled “Research on the Impact of Monitor Unit Optimization in Volumetric Modulated Arc Therapy Planning for Nasopharyngeal Carcinoma” (ID: PONE-D-25-12726).Those comments are all valuable and very helpful for revising and improving our paper, as well as the important guiding significance to our researches. We have studied comments carefully and have made correction which we hope meet with approval.

Revised portion are marked in red in the paper. The main corrections in the paper and the responds to the reviewer’s comments are as fllowing:

Responds to the reviewer’s comments:

Reviewer #1:

1. -Language and Style: While the manuscript is generally intelligible, some grammatical issues, tense inconsistencies, and awkward phrasing could be improved for clarity and flow. A light language edit by a native English speaker or language service would help.

This document has undergone comprehensive grammatical refinement and tense optimization, with final polishing conducted by professional English language editors.

2. -Discussion Depth: The discussion could benefit from a slightly more in-depth comparison with existing studies that evaluated Strength variation in MUO parameters—notably the clinical implications of underdosing in S100 could be expanded.

In the Discussion section, we have added comparative analyses with existing studies and further elaborated on the clinical implications of underdosage in Plan S100.

3. -Figures and Tables: The legends for Figures 1–5 could be made more detailed. Also, a summary figure or schematic explaining the MUO tool settings and their clinical impact would enhance readability.

The captions of Figures 1 to 5 have been supplemented with additional details. Additionally, a schematic diagram of the MUO tool's parameter settings has been added.

Reviewer #2:

1. In the manuscript draft section, the Author mentioned that the patient's data used in the

study was treated between October 23 to March 24. In contrast, the main abstract,

manuscript, and ethics statement were mentioned as October 2020 to March 2021.

The patient data used in this study were collected from those treated between October 2020 and March 2021, not October 2023 to March 2024. The manuscript draft has been revised accordingly and resubmitted.

2. In the main manuscript, under the heading 'VMAT planning,' how did the author decide to take the maximum MU 30 %?, Can it be more elaborate?

The maximum MU constraint was set at 30% of the base plan's total MU based on prior studies, which has now been referenced in the manuscript.

3. Oar is written in small letters. It must be in capital letters.

As requested, "Oar" has been corrected to "OAR" in all sections.

4. Under the heading “Comparison of the target and OAR doses….” The sentence

“excluding setup time and gantry rotation time….” Can be written as excluding the setup time and gantry repositioning time, or beam mode up time.

As requested, the “gantry rotation time” has been written as “beam mode up time”.

5. Under the heading “ pretreatment plan verification,” 10% threshold in the 3% 3mm

criteria? I doubt it is given in the cited reference 24.

The literature citations have now been corrected and properly referenced throughout the manuscript.

6. Table 4 can be rearranged.

Table 4 has been rearranged to improve clarity and logical flow of the presented data.

7. In Tables 5 and 6, what are t1, t2, and t3? The author must mention it and elaborate on these terms.

The t1, t2, and t3 values presented in Tables 5 and 6 have been clearly defined in the corresponding figure captions to ensure proper interpretation.

Reviewer #3:

1. Clarify whether MUO was applied during initial optimization or as a post-hoc re-optimization. This affects reproducibility and clinical implementation.

The MUO was implemented during the initial optimization phase before subsequent refinements.

2. Specify if OAR constraints were adjusted during MUO re-optimization. The observed increases in brainstem/spinal cord Dmax (≤1.5%) suggest constraints may need tightening.

During MUO re-optimization, the original constraints for both target and OARs remained unchanged, with adjustments applied solely to monitor unit optimization.

3. Justify the choice of gamma criteria (1%/1 mm, 2%/2 mm, 3%/3 mm). While 1%/1 mm is stringent, its clinical relevance should be discussed (e.g., is 3%/2 mm more practical for routine QA?).

The clinical significance of applying 1%/1mm gamma criteria was further elaborated, highlighting its stringent sensitivity in detecting teep dose gradients near critical structures compared to more lenient criteria.

4. For OARs, discuss if the increases in brainstem Dmax (≤1.5%) or thyroid V40 are acceptable given clinical tolerances.

While the re-optimized plans demonstrated modest increases in brainstem Dmax and thyroid V40, all values remained well within institutional clinical tolerance limits. It has been supplemented in the discussion.

5. Highlight that S100’ MU reduction (30.9%) comes at the cost of target underdosing in 8/21 patients, making S50 or S80 more viable.

The discussion has been revised to emphasize that compared to S100, S50 or S80 may represent more feasible compromise solutions.

6. Explain why MU reduction did not shorten delivery time (e.g., fixed gantry speed, dose rate limitations).

Additional discussion has been incorporated to address why MU reduction did not shorten delivery time.

7. Fixed MUO parameters (e.g., Maximum MU = 30% of base plan) may not be optimal for all cases. It is suggested to increase the discussion on the influence of the Maximum MU parameter on the results.

The impact of the Maximum MU parameter on the planning results has been supplemented in the discussion.

8.Suggest prospective validation with larger cohorts, long-term toxicity outcomes, and exploration of adaptive MUO strategies.

Future studies will include prospective validation with more cases and long-term toxicity outcome assessments.

9.In the Abstract, emphasize the improvement in gamma passing rates with MU reduction, as this is a notable practical outcome.

In the abstract, key emphasis has been placed: the significant improvement in Gamma passing rate achieved by reducing MU (Monitor Units), a result with important clinical significance.

10.Ensure consistent formatting (e.g., italicize journal names, use "et al." correctly). The citation formats are inconsistent. Some authors use their full names while others use abbreviations.

The manuscript formatting has been carefully reviewed, and the reference style has been corrected accordingly.

---

## [Editor Report · Decision Letter 1]

Impact of Monitor Unit Optimization in Volumetric Modulated Arc Therapy Planning for Nasopharyngeal Carcinoma

PONE-D-25-12726R1

Dear Dr. zeng,

We’re pleased to inform you that your manuscript has been judged scientifically suitable for publication and will be formally accepted for publication once it meets all outstanding technical requirements.

Kind regards,

Chung-Ta Chang

Academic Editor

PLOS ONE
---

## [Editor Report · Acceptance letter]

PONE-D-25-12726R1

PLOS ONE

Dear Dr. Zeng,

I'm pleased to inform you that your manuscript has been deemed suitable for publication in PLOS ONE. Congratulations! Your manuscript is now being handed over to our production team.

Kind regards,

on behalf of

Dr. Chung-Ta Chang

Academic Editor

PLOS ONE